# Comparative Assessment of Functional and Morphological Markers in Guinea Pig (*Cavia porcellus*) Oocytes Collected at Different Estrous Cycle Phases

**DOI:** 10.3390/ani15131953

**Published:** 2025-07-02

**Authors:** Jorge X. Samaniego, José L. Pesántez, Fernando P. Perea, Andrea P. Pazmiño, Jorge B. Dután, Salvador Ruiz

**Affiliations:** 1Faculty of Agriculture Sciences, University of Cuenca, Cuenca 010107, Ecuador; 2Department of Physiology, Faculty of Veterinary, University of Murcia, 30100 Murcia, Spain; 3Institute for Biomedical Research of Murcia, IMIB-Arrixaca, 30120 Murcia, Spain

**Keywords:** *Cavia porcellus*, oocyte maturation, cytoplasmic maturation, apoptosis detection, lipid droplets, Brilliant Cresyl Blue staining

## Abstract

Reproductive technologies offer valuable tools to enhance both animal production and biomedical research. However, their success is critically dependent on the selection of high-quality oocytes. In guinea pigs, the application of these technologies has yielded limited efficiency, potentially due to the intrinsic oocyte characteristics. This study aimed to assess oocyte quality in female guinea pigs at two distinct stages of the reproductive cycle in order to identify the optimal timing for oocyte retrieval. Morphological and cytological parameters were assessed, including oocyte diameter, structural integrity, cytoplasmic composition, and cellular viability. Oocytes obtained during the diestrus stage exhibited greater diameter and improved cytoplasmic organization, suggestive of enhanced developmental competence. Conversely, oocytes retrieved during the periovulatory stage—closer to the time of natural ovulation—showed increased nuclear maturity and fewer indicators of early cellular degeneration. These findings indicate that the optimal stage for oocyte collection may vary depending on the specific attributes prioritized in in vitro applications. A deeper understanding of these stage-dependent differences may assist researchers and clinicians in selecting the most appropriated collection window, thereby improving the efficiency of embryo production protocols and contributing to advances in both animal husbandry and medical research.

## 1. Introduction

The guinea pig (*Cavia porcellus*) is a hystricomorph rodent domesticated over 3000 years ago, with growing importance in both animal production and biomedical research [1,2]. In addition to its efficient feed conversion and nutritional relevance in Andean countries [3,4], this species serves as a valuable model for physiological and reproductive research due to unique features such as placental development and a prolonged estrous cycle, which more closely resemble those of primates than other rodents [5,6]. However, its reproductive physiology poses several challenges to the advancement of assisted reproductive technologies (ART), including small litter sizes, difficulties in estrus synchronization, and limited ovarian response to superovulation protocols [7,8].

Despite substantial progress in other domestic species, where in vitro maturation (IVM), fertilization, and embryo culture systems are standardized and yield consistent results [9,10], IVM outcomes in guinea pigs remain highly variable [11]. Factors such as poor oocyte quality and asynchrony between nuclear and cytoplasmic maturation have been identified as potential contributors to this inefficiency [8,12]. In a previous study, we evaluated the effect of follicular fluid (egpFF) and serum (egpS) from estrous females on oocyte maturation, observing improvements in cumulus cell expansion and nuclear progression, particularly in oocytes of superior morphological quality [13]. However, overall maturation rates remained low, suggesting the presence of intrinsic physiological limitations that hinder oocyte developmental competence in this species [8,11,12].

One critical yet understudied factor influencing oocyte quality in guinea pigs is the stage of the estrous cycle at the time of collection. In several species, oocyte competence varies significantly across reproductive stages due to underlying endocrine and metabolic fluctuations [14,15]. Specifically, the diestrus and periovulatory stages present distinct hormonal profiles and follicular dynamics that are likely to impact oocyte characteristics such as size, lipid content, metabolic activity, and susceptibility to apoptosis [16,17]. However, to date, no studies have systematically compared these parameters in guinea pigs, leaving a significant knowledge gap that limits the optimization of IVM protocols [11].

In this context, the present study aimed to evaluate and compare morphological and functional markers of oocyte quality in guinea pigs during the diestrus and periovulatory stages, with the goals of elucidating potential physiological determinants contributing to the previously reported low maturation rates and identifying the optimal stage for harvesting oocytes with high developmental potential. To this end, we employed a comprehensive analytical approach that included assessment of G6PD activity using the BCB test, evaluation of cytoplasmic lipid distribution, quantification of early apoptotic changes, measurement of oocyte diameter, and analysis of nuclear progression following in vitro culturing. This integrated characterization is expected to support the development of more precise criteria for selecting developmentally competent oocytes, thereby improving the efficiency and reproducibility of IVM protocols in this physiologically distinct species.

## 2. Materials and Methods

### 2.1. Chemicals and Media

Unless otherwise stated, all reagents and chemicals were purchased from Sigma (St. Louis, MO, USA). LipidSpot™ and the Apoptotic, Necrotic, and Healthy Cells Quantification Kit Plus were sourced from Biotium (Fremont, CA, USA). Folltropin-V was obtained from Vetoquinol (Alcobendas, Madrid, Spain) and altrenogest (Regumate^®^) from MSD Animal Health (Igoville, Normandy, France).

### 2.2. Animals and Synchronization Protocol

A total of 20 healthy, cycling female guinea pigs, aged 3–9 months and weighing 600–800 g, were obtained from a certified commercial breeding and processing farm. Animals younger than 3 months were not considered because the early post-pubertal period in guinea pigs is often characterized by irregular estrous cycles, variable ovarian activity, and a higher incidence of anovulatory cycles. As part of the experimental design, the animals underwent an oral estrous synchronization protocol with altrenogest (0.22 mg/kg/day) for 15 consecutive days, following the protocol described by Grégoire et al. [6]. This non-invasive treatment was administered by trained farm personnel under the direction of the research team.

Following synchronization, estrous stage was determined via external vaginal membrane assessment, and the animals were assigned to two groups corresponding to the diestrus (DIG) and periovulatory (POG) phases. The animals were humanely slaughtered as part of routine farm operations, and ovaries were collected postmortem for oocyte recovery and analysis. All procedures were conducted in accordance with local regulations on animal welfare and humane slaughter practices. A detailed statement on ethical compliance, including the post-mortem nature of sample collection and approval by the institutional review board, is provided in the “Institutional Review Board Statement” section of this manuscript.

### 2.3. Oocyte Collection and Classification

Two (Day 17) and ten (Day 25) days after progestogen withdrawal, the animals were euthanized by intraperitoneal administration of sodium pentobarbital at a dose equivalent to twice the recommended anesthetic dose for guinea pigs (i.e., 45 to 70 mg/kg). The ovaries were immediately removed and transported to the laboratory in pre-warmed Ringer’s lactate solution maintained at a physiological temperature of 35–37 °C. Upon arrival, the ovaries were washed three times, and cumulus–oocyte complexes (COCs) were retrieved by repeatedly slicing the ovarian cortex with sterile surgical blades.

The collected COCs were classified under a stereomicroscope (Nikon, Tokyo, Japan) according to morphological criteria adapted from Wang et al. [8] and Yao et al. [12]. The classification system included multiple categories based on the number of cumulus cell layers and ooplasm homogeneity, as summarized in Table 1. This classification was applied to the cumulus–oocyte complexes (COCs) as intact structures, rather than to denuded oocytes in isolation.

A total of 744 oocytes were collected postmortem from the 20 synchronized females, yielding an average of approximately 37 oocytes per animal. Following morphological classification, the oocytes were allocated into two experimental groups. The first group (*n* = 332) was employed to assess metabolic activity via Brilliant Cresyl Blue (BCB) staining and cytoplasmic lipid distribution using Nile Red staining. The second group (*n* = 412) was utilized to determine apoptotic status (Annexin V and ethidium bromide), nuclear maturation stage (Hoechst 33342), and oocyte diameter.

### 2.4. Brilliant Cresyl Blue Test

The Brilliant Cresyl Blue (BCB) test is employed to assess the activity of the enzyme glucose-6-phosphate dehydrogenase (G6PD), which remains active in growing, immature oocytes [18]. Since G6PD activity decreases as the oocytes complete their growth, the BCB test serves as an indirect marker of oocyte developmental competence. Oocytes were incubated in a 26 μM BCB solution for 90 min at 38.5 °C. Following incubation, they were washed three times in phosphate-buffered saline (PBS) to remove excess dye and then assessed under a stereomicroscope. Oocytes that retained the blue stain (BCB+) exhibited reduced G6PD activity, suggesting that they had completed the growth phase and were potentially more developmentally competent. Conversely, oocytes that did not retain the blue stain (BCB−) showed high G6PD activity, indicative of continued growth and lower competence [19].

### 2.5. Lipid Distribution Patterns

To evaluate cytoplasmic lipid distribution patterns, the fluorogenic neutral lipid stain LipidSpot™ 488 (Biotium, #70065) was used, following the manufacturer’s protocol for live-cell staining. Oocytes previously assessed for G6PDH activity were denuded using hyaluronidase, washed with PBS, and fixed in 4% formaldehyde for 30 min at room temperature. After fixation, oocytes were thoroughly rinsed in PBS and stained with LipidSpot™ 488 for 10 min in the dark. Subsequently, they were mounted on microscope slides using glycerol as a mounting medium and observed under an Eclipse Ci epifluorescence microscope (Nikon, Tokyo, Japan) equipped with appropriate excitation and emission filters.

Oocytes were classified according to lipid distribution patterns in the cytoplasm as follows: “HLD”—homogeneous lipid distribution throughout the cytoplasm, and “CLD”—centralized lipid distribution located in the central region of the cytoplasm.

### 2.6. In Vitro Maturation (IVM) of COCs

The IVM medium was prepared according to the protocol described by Samaniego et al. [13], consisting of TCM-199 with Earle’s salts, supplemented with 10% fetal bovine serum (FBS), 0.2 mM sodium pyruvate, 25 μg/mL follicle-stimulating hormone (FSH), 5 μg/mL luteinizing hormone (LH), 30 ng/mL epidermal growth factor (EGF), 50 μg/mL gentamicin sulfate, and 5 μg/mL estradiol-17β. The culture was carried out at 38.5 °C in a humidified atmosphere with 5% CO_2_ for 24 h. Each group of COCs (classified as Type A, B, or C) was washed twice in the maturation medium. IVM was conducted in 70 μL drops, with 15 oocytes per drop.

### 2.7. Apoptosis Assessment

Apoptosis in COCs was assessed using the Apoptotic, Necrotic, and Healthy Cells Quantification Kit (Biotium, #30018), following the manufacturer’s instructions. After IVM, oocytes were denuded using hyaluronidase, washed with PBS, and subsequently fixed in 4% formaldehyde for 30 min at room temperature. Following fixation, oocytes were thoroughly rinsed and stained with the reagents provided in the kit for 15 min in the dark. After staining, the oocytes were washed with the 1X Binding Buffer supplied by the manufacturer. Finally, the oocytes were mounted on microscope slides using glycerol and observed under an epifluorescence microscope equipped with a green filter. Based on the fluorescence staining patterns, oocytes were classified into three categories, as summarized in Table 2.

### 2.8. Nuclear Progression Assessment

COCs were denuded of cumulus cells by gentle pipetting in the presence of hyaluronidase and subsequently fixed in 4% formaldehyde in PBS for 30 min at room temperature. After fixation, the oocytes were thoroughly washed and stained with Hoechst 33342 (1 mg/mL) for 15 min in the dark at room temperature. The mounting medium consisted of PBS with glycerol and Hoechst 33342; slides were sealed with nail polish to prevent evaporation and preserve sample integrity. Oocytes were evaluated under an epifluorescence microscope at 200× and 400× magnifications. Nuclear morphology was used to classify oocytes into the following categories:Immature: Germinal vesicle (GV) present or oocyte in Metaphase I (MI).Mature—Metaphase II (MII): First polar body visible and metaphase present.Degenerated (DEG): Absence of discernible nuclear structures or evidence of chromatin fragmentation.

This classification provides a reliable assessment of meiotic progression and oocyte integrity following IVM.

### 2.9. Determination of Oocyte Diameter

Oocyte diameter was measured following staining with the Apoptotic, Necrotic, and Healthy Cells Quantification Kit. Oocytes were examined using an MS60 digital camera (MSHOT, Shenzhen, China) mounted on an epifluorescence microscope. Measurements were taken at the widest horizontal axis of each oocyte using ImageJ software (Java V 1.8.0), calibrated at 200× and 400× magnifications. Only intact, non-fragmented oocytes were included in the analysis to avoid measurement artefacts.

### 2.10. Statistical Analysis

All statistical analyses were performed using SPSS^®^ v.25 (IBM Corp., Armonk, NY, USA). The normality of quantitative variables was assessed using the Kolmogorov–Smirnov test. As the data did not follow a normal distribution, numerical values were log-transformed (base 10) prior to conducting parametric analyses. To evaluate the effect of the estrous cycle stage (periovulatory and diestrus) and the COC classification (Type A, B, and C) on oocyte diameter, a two-way analysis of variance (two-factor ANOVA) was conducted. Post hoc comparisons were carried out using Tukey’s test to identify significant differences between groups.

Logistic regression models were used to analyze variables associated with oocyte quality, with estrous cycle stage and COC classification included as fixed factors. A binary logistic regression model was applied to assess oocyte metabolic activity, based on Brilliant Cresyl Blue (BCB) staining (positive or negative) and lipid distribution patterns (homogeneous and centralized). Nuclear maturation status (immature, mature, degenerated) and apoptotic status (viable non-apoptotic, early apoptotic, late apoptotic oocytes) were analyzed using a multinomial logistic regression. The models estimated odds ratios (ORs) and 95% confidence intervals (CIs) for each predictor. Results were expressed as the mean ± SEM and percentages, and statistical significance was set at *p* < 0.05, while *p*-values of 0.05 < *p* < 0.1 were considered to show a trend.

## 3. Results

### 3.1. Evaluation of Brilliant Cresyl Blue Staining

COCs were classified as either BCB− or BCB+ based on their staining patterns following exposure to Brilliant Cresyl Blue (BCB) staining (Figure 1A). No significant interaction was observed between the estrous cycle stage and oocyte type in terms of the proportion of BCB− and BCB+ oocytes (*p* > 0.05). However, the percentage of BCB+ oocytes was significantly higher (*p* = 0.03) during the periovulatory stage compared to the diestrus stage (48.8% vs. 36.8%; Figure 1B). In contrast, COC classification had no significant effect on the proportion of BCB+ or BCB− oocytes (*p* = 0.612; Figure 1C).

### 3.2. Lipid Distribution Patterns in Guinea Pig Oocytes

No significant interaction was observed between the estrous cycle stage and the type of COC from which the oocytes were derived in relation to lipid distribution patterns (*p* > 0.05). Oocytes collected during the periovulatory stage exhibited a significantly lower proportion of uniform lipid distribution compared to those obtained during the diestrus stage (38.4% vs. 52.5%; *p* < 0.05). Conversely, the proportion of oocytes showing centralized lipid distribution was significantly higher in the periovulatory stage than in the diestrus stage (61.6% vs. 47.5%; *p* < 0.05; Figure 2). Representative fluorescence micrographs of oocytes stained to assess cytoplasmic lipid distribution are shown in Figure 3.

Oocytes from the periovulatory phase were 2.42 times more likely to exhibit a central lipid distribution compared to those from the diestrus phase (OR: 2.419; 95%CI: 1.167–5.013; *p* = 0.018).

Additionally, a statistical trend was observed (*p* = 0.059), suggesting a relationship between cytoplasmic lipid distribution patterns and COC classification. Oocytes from Type A COCs exhibited a significantly higher proportion of homogeneous lipid distribution (71.4%) compared to those from Type B and Type C COCs, which showed similar and lower percentages (45.5% and 43.8%, respectively; *p* < 0.05; Figure 2). Likewise, oocytes from Type A COCs showed a lower proportion of centralized lipid distribution compared to those from Type B and Type C COCs, which exhibited higher and comparable proportions (28.6% vs. 54.6% and 56.3%, respectively; Figure 2).

### 3.3. Apoptotic Status

No significant interaction was found between the estrous cycle stage and the type of COC from which the oocytes were derived in relation to oocyte apoptotic status (*p* > 0.05). Nevertheless, the distribution of apoptotic states differed significantly between estrous cycle stages (Figure 4; *p* = 0.014). Figure 5 shows representative fluorescence images of oocytes classified according to apoptotic status using Annexin V and ethidium bromide staining. The percentages of viable (non-apoptotic) oocytes were similar in both stages (periovulatory: 16.9%; diestrus: 11.0%). In contrast, the proportion of oocytes in early apoptosis was significantly lower during the periovulatory stage compared to the diestrus stage (14.6% vs. 25.5%; *p* < 0.05; Figure 4). No significant differences were observed in the percentages of oocytes in late apoptosis between the two phases (periovulatory: 68.5; diestrus: 63.4%). Interestingly, the combined proportion of oocytes in early and late apoptosis remained relatively high regardless of the estrous cycle stage. When analyzed by COC classification, oocytes from Type A, B, and C COCs showed no statistically significant differences in apoptotic status (Figure 4B).

### 3.4. Nuclear Maturation

During the evaluation of nuclear maturation in guinea pig oocytes, a significant interaction was observed between the estrous cycle stage and the type of COC from which the oocytes were derived (*p* = 0.001). The percentage of mature oocytes was similar among the three COC types during the periovulatory stage (*p* > 0.05). However, during the diestrus stage, oocytes from Type A COCs exhibited a significantly higher percentage of maturation (30.0%; *p* < 0.05; Figure 6) compared to oocytes from Type B and Type C COCs, which showed lower and comparable values (5.7% and 3.4%, respectively; *p* > 0.05).

The proportion of immature oocytes was significantly higher in oocytes from Type A COCs during both estrous cycle stages (periovulatory: 80.0%; diestrus: 70.0%; *p* < 0.05), in comparison to oocytes from Type B and Type C COCs, which showed similar percentages of immaturity across both stages. In the periovulatory stage, the proportion of immature oocytes was 69.9% for oocytes from Type B COCs and 53.3% for those from Type C COCs, whereas in the diestrus stage, these values were 69.8% and 37.9%, respectively. These results suggest a higher likelihood of nuclear immaturity during the periovulatory stage (OR = 2.04; *p* = 0.011), particularly in oocytes from Type A COCs (OR = 5.36; *p* = 0.010) and from Type B COCs (OR = 3.35; *p* < 0.001). Figure 7 shows fluorescence images of oocytes at various nuclear stages after Hoechst 33342 staining, including GV, MI, MII, and degenerated stages.

The percentage of degenerated oocytes was highest among oocytes from Type C COCs during both estrous stages (periovulatory: 30.0%; diestrus: 58.6%), in contrast to oocytes from Type A and Type B COCs, which showed lower and similar values (*p* > 0.05). Specifically, during the periovulatory stage, the proportions of degenerated oocytes were 20.0% for Type A and 13.3% for Type B COCs, while during the diestrus stage, the values were 0.0% for Type A and 24.5% for Type B COCs (Figure 6). Notably, no degenerated oocytes were observed among those from Type A COCs in the diestrus phase, and no oocytes at the MII stage were observed among Type A COCs during the periovulatory phase. These absences are reflected in the graph where no bars are shown for these specific combinations.

### 3.5. Evaluation of Morphology in Guinea Pig Oocytes

Oocyte diameter in guinea pigs was significantly influenced by both the estrous cycle stage and the oocyte type of the COC from which the oocytes were derived (*p* = 0.03). Oocytes from Type A and Type C COCs collected during the diestrus stage exhibited significantly greater diameter compared to those obtained from the same COC types during the periovulatory stage. In contrast, no significant difference in diameter was observed in oocytes from Type B COCs between estrous stages (*p* > 0.05; Figure 8).

## 4. Discussion

This study provides novel insights into the morphological and functional quality of guinea pig oocytes collected at different stages of the estrous cycle. Our findings demonstrate that both the estrous cycle phase and COC classification significantly influence key parameters associated with oocyte competence, including G6PD activity, cytoplasmic lipid distribution, apoptotic status, and nuclear maturation.

BCB test results revealed that, although the distribution of BCB+ and BCB− oocytes was not affected by COC classification, the proportion of BCB+ oocytes was markedly higher during the periovulatory phase. This suggests that the metabolic transition associated with reduced G6PD activity—a biochemical marker indicative of oocyte growth completion—is more prevalent during this reproductive stage, reflecting a more advanced cytoplasmic maturation profile. These findings align with previous reports linking the periovulatory phase to enhanced oocyte competence [19,20]. Furthermore, studies in rabbits have demonstrated that BCB+ oocytes not only exhibit higher in vitro maturation and developmental potential, but also lower levels of apoptosis compared to their BCB− counterparts [21], supporting the reliability of G6PD activity as a predictor of functional viability.

Cytoplasmic lipid distribution, a recognized marker of cytoplasmic maturation, also exhibited distinct distribution in relation to both estrous phase and COC morphology. Homogeneous lipid distribution was more frequently observed in oocytes derived from Type A COCs and those collected during the diestrus phase. In contrast, centralized lipid distribution, often considered indicative of cytoplasmic immaturity or cellular stress, was more frequently observed in oocytes from Type B and Type C COCs, as well as in those obtained during the periovulatory phase. Although the association between lipid pattern and COC classification reached only a statistical trend, the biological pattern suggests that developmental competence may be influenced by both intrinsic follicular characteristics and the surrounding physiological environment. These findings challenge the prevailing notion that the periovulatory phase universally yields oocytes with greater developmental potential. Instead, they suggest that the diestrus phase, particularly in oocytes derived from Type A COCs, may provide more favorable cytoplasmic conditions for IVM.

Previous studies in porcine models have demonstrated that lipid distribution within oocyte cytoplasm is closely linked to oocyte developmental competence, with homogeneous patterns associated with improved nuclear maturation, fertilization capacity, and embryonic development rates [22,23]. Considering these parallels, our data suggest that lipid droplet organization in guinea pig oocytes may serve as a valuable proxy for cytoplasmic maturation status and should be integrated into future protocols for oocyte selection and quality assessment.

Apoptotic analysis revealed high rates of both early and late apoptosis across all groups; however, the proportion of early apoptotic oocytes was significantly lower during the periovulatory phase. These findings may suggest a transient window of improved cellular viability during this phase. Nevertheless, the consistently elevated apoptosis rates underscore the need to optimize handling protocols and in vitro culture conditions. Apoptotic status was not associated with COC morphological classification, but rather with the estrous cycle phase. This supports the hypothesis that hormonal dynamics and follicular environment at the time of oocyte collection have a stronger impact on apoptotic susceptibility than morphological quality alone [24].

Apoptosis is a tightly regulated process that plays a central role in follicular atresia, and its activation in oocytes may compromise meiotic progression, chromatin integrity, and organelle function. Elevated apoptosis in oocytes selected for IVM may therefore be detrimental to subsequent fertilization and embryo development. Strategies aimed at reducing apoptotic activation—such as antioxidant supplementation, anti-apoptotic agents, or optimized culture media—should be explored in future studies to improve oocyte survival and developmental potential.

Nuclear maturation analysis revealed complex interactions. During the diestrus phase, oocytes derived from Type A COCs showed the highest percentage of maturation, while those from Type B and Type C COCs exhibited limited progression. In contrast, during the periovulatory phase, maturation rates were more uniform across COC types, although still relatively low. These results suggest that intrinsic oocyte quality—reflected by COC morphology—is a key determinant of nuclear competence, particularly during diestrus. These observations align with prior research indicating that cytoplasmic and nuclear maturation processes must be synchronized for successful fertilization and embryo development [25,26]. Moreover, our earlier findings support this interpretation, as in vitro maturation of guinea pig oocytes supplemented with estrous guinea pig serum (egpS) or follicular fluid (egpFF) similarly resulted in low maturation rates, regardless of the protein supplement used. This reinforces the notion that impaired nuclear maturation may not be solely attributable to suboptimal culture conditions but may also reflect inadequate oocyte selection and/or suboptimal timing of collection relative to the estrous cycle [13].

A significant interaction was observed between estrous phase and COC classification with respect to oocyte diameter. Interestingly, oocytes from Type A and Type C COCs collected during diestrus showed greater diameters than those collected during the periovulatory phase. This difference may reflect a longer duration of follicular growth and greater cytoplasmic accumulation in the hormonal context of diestrus. Diameter has been proposed as a non-invasive marker of oocyte competence, as it is associated with mitochondrial maturation, ATP reserves, and organelle redistribution [27]. The lack of diameter variation among oocytes from Type B COCs suggests greater heterogeneity or impaired growth within this group.

Together, these findings emphasize the multifactorial nature of oocyte quality. Both intrinsic factors (COC morphology, diameter, cytoplasmic traits) and extrinsic factors (estrous phase, endocrine environment) act in concert to modulate developmental competence. Selecting oocytes based on integrated morphological and physiological criteria, including COC classification, cytoplasmic lipid patterns, and BCB response, may significantly enhance IVM outcomes. This multifaceted approach could serve as a foundation for optimizing in vitro embryo production protocols in guinea pigs and potentially in other species with similar reproductive challenges.

## 5. Conclusions

The present study demonstrates that both the estrous cycle phase and oocyte morphological classification significantly affect the morphological and functional quality of guinea pig oocytes. Oocytes collected during the diestrus phase, particularly those classified as type A, exhibited larger diameters and more favorable lipid distribution patterns, indicative of enhanced cytoplasmic maturity. In contrast, oocytes obtained during the periovulatory phase showed greater metabolic competence and lower rates of early apoptosis.

These findings highlight the multifactorial nature of oocyte competence, in which both intrinsic factors (e.g., oocyte type) and extrinsic conditions (e.g., cycle stage) play critical roles. The integration of markers such as G6PD activity (via BCB staining), lipid distribution, apoptosis status, and nuclear maturation provides a comprehensive framework for evaluating oocyte quality.

Optimization of oocyte selection based on these criteria may improve the efficiency of in vitro maturation and embryo production protocols in *Cavia porcellus*. Future research should focus on combining these markers with hormonal profiling and fertilization outcomes to further refine assisted reproductive techniques in this established biomedical and agricultural model.

## Figures and Tables

**Figure 1 animals-15-01953-f001:**
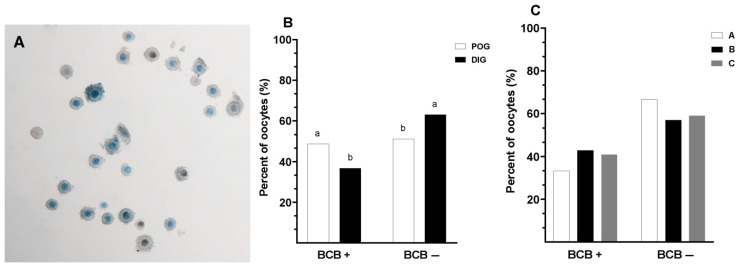
Comparison of Brilliant Cresyl Blue (BCB− and BCB+) cumulus–oocyte complexes (COCs) in guinea pigs collected during two natural stages of the estrous cycle. (**A**) Representative images of BCB staining in COCs, where differences in cytoplasmic coloration reflect the oocyte metabolic activity. Oocytes with a colorless (unstained) ooplasm were classified as BCB−, indicating high glucose-6-phosphate dehydrogenase (G6PDH) activity and thus immature or metabolically active oocytes. In contrast, oocytes with a blue-stained ooplasm were classified as BCB+, denoting reduced G6PDH activity typically associated with more developmentally competent oocytes. (**B**) Percentage of BCB+ and BBC- oocytes by estrous stage: periovulatory (POG) and diestrus (DIG). (**C**) Distribution of BCB staining results according to COC classification (Type A, B, and C COCs). Data are presented as percentages. Different lowercase letters (a–b) indicate significant differences between groups (*p* < 0.05).

**Figure 2 animals-15-01953-f002:**
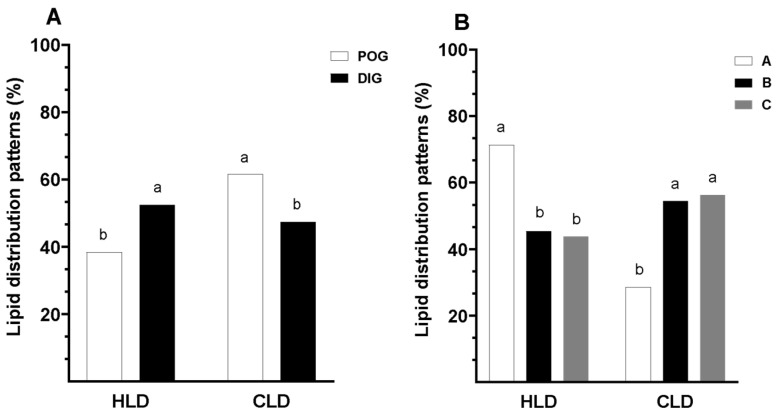
Assessment of oocyte lipid distribution patterns in guinea pigs collected during two natural stages of the estrous cycle. (**A**) Distribution of lipid patterns by estrous cycle stage: periovulatory (POG) and diestrus (DIG). (**B**) Distribution according to COC classification (Type A, B, and C COCs). Lipid patterns were classified as homogeneous lipid distribution (HLD) and centralized lipid distribution (CLD). Data are presented as percentages. Different lowercase letters (a–b) indicate significant differences between groups (*p* < 0.05).

**Figure 3 animals-15-01953-f003:**
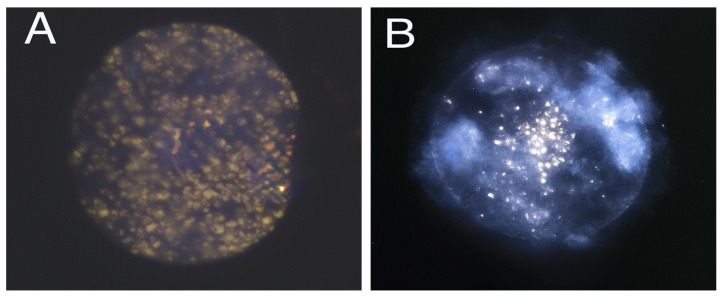
Representative images of lipid distribution patterns in stained guinea pig oocytes. (**A**) “HLD”: homogeneous lipid distribution throughout the cytoplasm, and (**B**) “CLD”: centralized lipid distribution located in the central region of the cytoplasm.

**Figure 4 animals-15-01953-f004:**
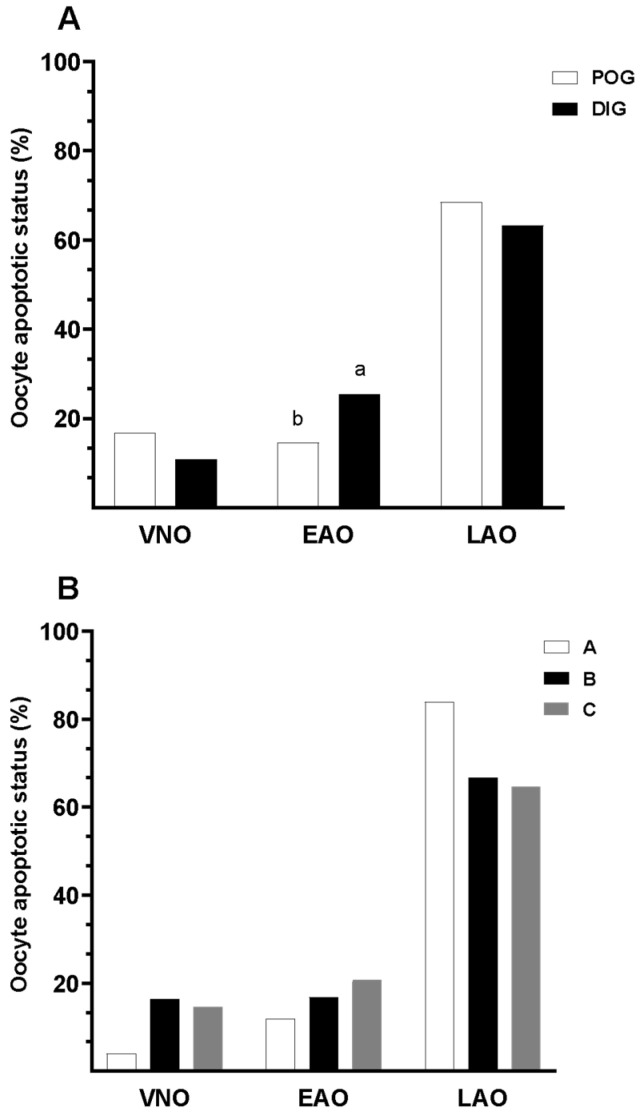
Detection of oocyte apoptotic status in guinea pigs collected during two natural estrous cycle stages. (**A**) Distribution of apoptotic categories by estrous stage: periovulatory (POG) and diestrus (DIG). (**B**) Distribution according to COC classification (Type A, B, and C COCs). Apoptotic status was classified as follows: viable non-apoptotic oocyte (VNO); early apoptotic oocyte (EAO) and late apoptotic oocyte (LAO). Data are presented as percentages. Different lowercase letters (a–b) indicate significant differences between groups (*p* < 0.05).

**Figure 5 animals-15-01953-f005:**
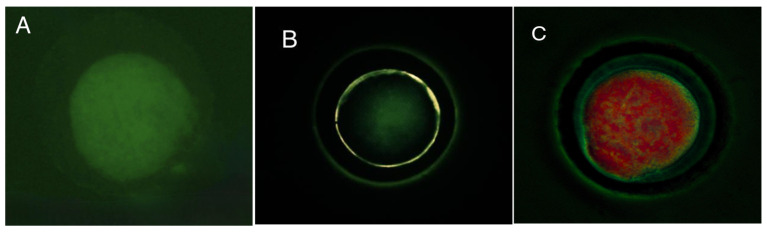
Representative images of oocytes classified according to apoptotic status: (**A**) viable non-apoptotic oocytes (VNO; negative for both Annexin V and ethidium bromide [EtBr], indicating intact membranes and no signs of apoptosis); (**B**) early apoptotic oocytes (EAO; Annexin V-positive, EtBr-negative, indicating the externalization of phosphatidylserine as an early marker of apoptosis while maintaining membrane integrity); and (**C**) late apoptotic oocytes (LAO; positive for both markers, indicating phosphatidylserine externalization and compromised membrane integrity associated with advanced apoptosis).

**Figure 6 animals-15-01953-f006:**
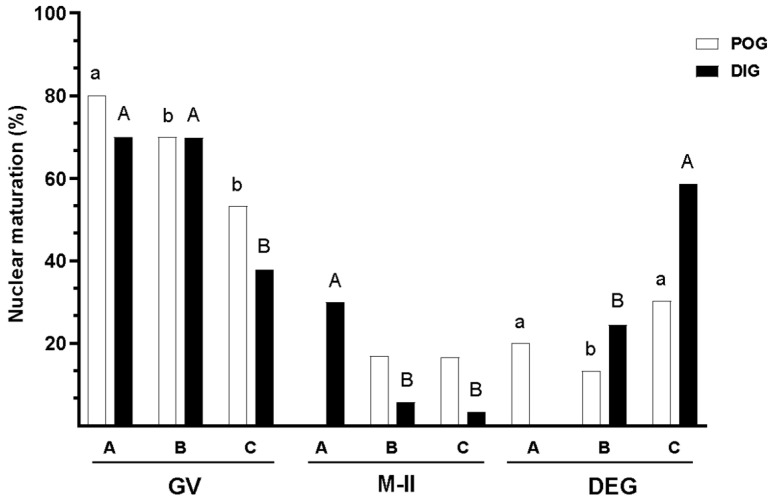
Evaluation of nuclear maturation in guinea pigs collected during two natural stages of the estrous cycle: periovulatory (POG) and diestrus (DIG). Oocytes were analyzed according to the classification of the COCs from which they were derived (Type A, B, and C COCs) and their nuclear maturation status, defined as follows: immature—germinal vesicle (GV); mature—metaphase II (M-II), and degenerated—DEG. Data are presented as percentages. Different lowercase letters (a–b) indicate significant differences between oocyte types within the periovulatory stage (*p* < 0.05), while different uppercase letters (A–B) indicate significant differences between oocyte types within the diestrus stage (*p* < 0.05). Bars are not shown for groups in which no oocytes were observed (*n* = 0).

**Figure 7 animals-15-01953-f007:**
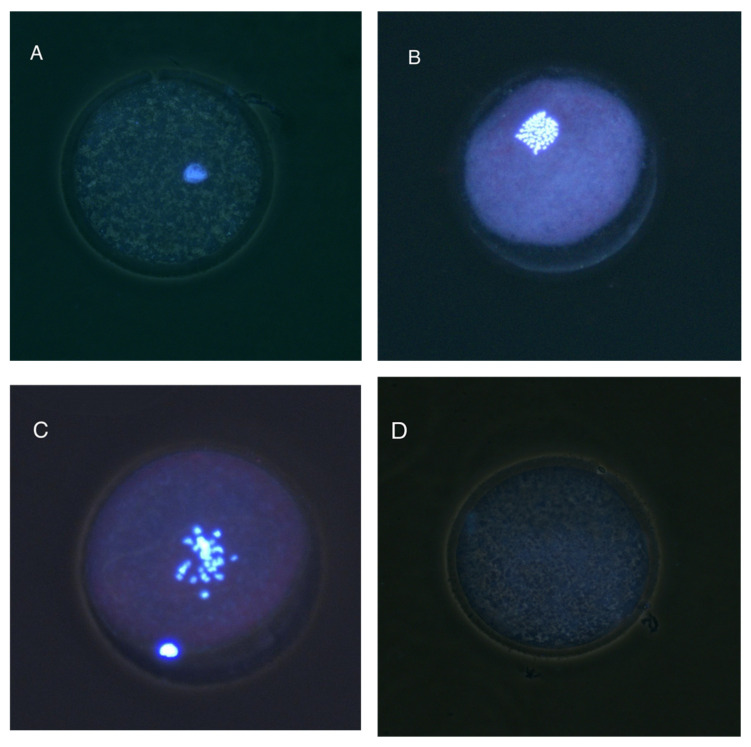
Representative images of nuclear maturation stages in guinea pig oocytes stained with Hoechst 33342. (**A**) Germinal vesicle (GV): Immature. (**B**) Metaphase I (MI): Immature. (**C**) Metaphase II (MII): Mature. (**D**) Degenerated oocyte: (DEG).

**Figure 8 animals-15-01953-f008:**
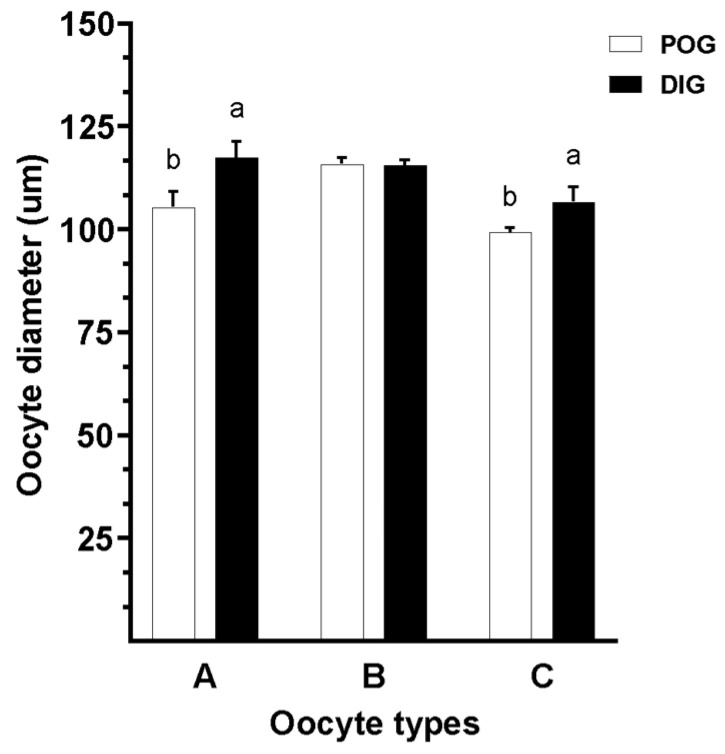
Evaluation of oocyte morphological diameter (µm) in guinea pigs collected during two natural stages of the estrous cycle: periovulatory (POG) and diestrus (DIG). Oocytes were grouped based on the COC classification (Type A, B, and C COCs) from which they were recovered. Data are represented as the mean ± S.E.M. A significant interaction was observed between estrous cycle stage and oocyte quality (*p* < 0.03). Different lowercase letters (a–b) indicate significant differences between groups (*p* < 0.05).

**Table 1 animals-15-01953-t001:** Classification of COCs based on morphological criteria.

Category	Description
A	Oocytes surrounded by four or more layers of cumulus cells.
B	Few layers of cumulus coating, homogeneous or irregular cytoplasm.
C	Expanded cumulus cell coating, partial or nonexistent.

Classification of COCs was performed according to cumulus cell investment and oocyte cytoplasmic appearance, based on morphological criteria described in previous studies.

**Table 2 animals-15-01953-t002:** Classification of oocytes based on Annexin V and ethidium bromide staining.

Classification	Annexin V (+)	Ethidium Bromide (+)	Interpretation
Viable Non-Apoptotic Oocyte (VNO)	−	−	Oocyte with no signs of apoptosis.
Early Apoptotic Oocyte (EAO)	+	−	Oocyte in early stages of apoptosis.
Late Apoptotic Oocyte (LAO)	+	+	Oocyte in late apoptosis or necrotic stage.

Annexin V binds specifically to phosphatidylserine residues that become exposed on the outer leaflet of the plasma membrane during early stages of apoptosis. This translocation is one of the earliest detectable events in the apoptotic cascade, occurring prior to the membrane rupture. Ethidium bromide is a DNA-intercalating fluorescent dye that penetrates only cells with compromised membranes, thus serving as a marker for late apoptosis or necrosis. The combination of these markers allows the classification of oocyte viability and apoptotic status.

## Data Availability

The data that support the findings of this study are available from the corresponding author upon reasonable request.

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
