# Peer review of "Comparative Assessment of Functional and Morphological Markers in Guinea Pig (*Cavia porcellus*) Oocytes Collected at Different Estrous Cycle Phases"

_animals, 2025, doi:10.3390/ani15131953_

Round 1

Reviewer 1 Report

Comments and Suggestions for Authors

The manuscript investigated morphological and functional markers of oocyte quality in guinea pigs during diestrus and the periovulatory stage. The goal was to elucidate potential physiological determinants that contribute to the previously reported low maturation rates and to identify the optimal stage for harvesting oocytes with high developmental potential. Overall, the manuscript is well-written and introduces new ideas to improve assisted reproduction in this species.

List of Corrections:

Simple summary

Lines 18 and 19; there are some typos to be revised. Ex.   ... assessed, , including.... oocyte ..... composition and, and cellular

Introduction:

Lines 67-73: I advise deleting this paragraph. Information about oocyte cytoplasmic and nuclear maturation is already included in the previous paragraph.  There is no need to add information about the techniques (BCB, etc.) in the introduction. Some data about them is included in the methodology.

Material and Methods

Lines 115-116: It is important to include information about the approval of this animal study by the institution's ethical committee.

Table 1 shows the classification of COCs as types A, B, or C; however, in the statistical analysis and results, the author states that the oocytes were classified as A, B, or C, which is confusing. I advise using "Type A COCs," "Type B COCs," and "Type C COCs" throughout the manuscript, including the figures.

Results:

Lines 254 and elsewhere: Replace "Type A oocytes" with "Oocytes from Type A COCs."

In Figures 1, 2, and 3, replace A, B, and C with "Type A COCs," "Type B COCs," and "Type C COCs."

For example, on line 302, it is confusing to say, "The proportion of immature oocytes was significantly higher in type A oocytes," when it should say, "The proportion of immature COCs was significantly higher in type A COCs."

Discussion

In general, the discussion needs improvement. There are many short paragraphs that discuss the same subjects, for example, lines 344–361 and 363–384. I advise joining them. Start a new paragraph only when you change the subject.

Line 456: There is no need for a paragraph about the limitations of this study.

Line 470: Remove the paragraph about the conclusion, it is repeated in the next topic.

Comments on the Quality of English Language

In general, the manuscript is well-written.

Author Response

Dear Reviewer,

We would like to sincerely thank you for your valuable comments and suggestions, which have greatly contributed to improving the quality and clarity of our manuscript.

We have carefully addressed each point raised in the review and provided detailed responses in the attached document.

Please do not hesitate to let us know if further clarification is needed.

Thank you once again for your thoughtful review.

Kind regards,
Jorge X. Samaniego
On behalf of all co-authors

Reviewer 2 Report

Comments and Suggestions for Authors

The proposed article compares the results of procedures for in vitro maturation of oocytes from guinea pigs in two stages of their estrous cycle – diestrus and preovulatory phase. The morphological characteristics of the cumulus-oocyte complexes, the degree of maturity and apoptosis of oocytes, as well as their metabolic activity have been studied.

I think that the article needs significant additions. My suggestions are:

To give the total number of oocytes and the number of oocytes per animal.

To describe the distribution of oocytes between different experiments.

Please, provide microscopic images of oocytes examined for Lipid Distribution Patterns.

Provide microscopic images of oocytes examined for Apoptotic Status should be provided.

Provide microscopic images of oocytes examined for Nuclear Maturation.

The article states that the observed non-degenerate oocytes are either mature (at metaphase II) or at the GV stage. No data was given for immature oocytes in metaphase I. They should also be reported and included in the statistics.

Author Response

(The authors gave the same response as above.)

Reviewer 3 Report

Comments and Suggestions for Authors

The article covers an important topic in physiology and embryology. Understanding the factors that affect the maturation of ovarian follicles and oocytes is crucial for grasping their significance in procedures involving oocyte retrieval and the production of in vitro embryos in both animals and humans. I ask the authors to clarify two points.
1. What was the reason for selecting individuals at three months of age? Were younger individuals, just after reaching sexual maturity, considered?
2. Figure 4 lacks two bars for type A oocytes: one for the periovulatory stage at maturity, and one for the diestrus stage at the DEG stage. Did the authors not observe these forms in these phases?

Author Response

(The authors gave the same response as above.)

Round 2

Reviewer 2 Report

Comments and Suggestions for Authors

The authors have complied with all proposals. I think that the article can be accepted for publication.